# FPNN: Field Probing Neural Networks for 3D Data

**Yangyan Li**[1,2]    **Sören Pirk**[1]    **Hao Su**[1]    **Charles R. Qi**[1]    **Leonidas J. Guibas**[1]

[1]Stanford University, USA          [2]Shandong University, China

## Abstract

Building discriminative representations for 3D data has been an important task in computer graphics and computer vision research. Convolutional Neural Networks (CNNs) have shown to operate on 2D images with great success for a variety of tasks. Lifting convolution operators to 3D (3DCNNs) seems like a plausible and promising next step. Unfortunately, the computational complexity of 3D CNNs grows cubically with respect to voxel resolution. Moreover, since most 3D geometry representations are boundary based, occupied regions do not increase proportionately with the size of the discretization, resulting in wasted computation. In this work, we represent 3D spaces as volumetric fields, and propose a novel design that employs field probing filters to efficiently extract features from them. Each field probing filter is a set of probing points — sensors that perceive the space. Our learning algorithm optimizes not only the weights associated with the probing points, but also their locations, which deforms the shape of the probing filters and adaptively distributes them in 3D space. The optimized probing points sense the 3D space "intelligently", rather than operating blindly over the entire domain. We show that field probing is significantly more efficient than 3DCNNs, while providing state-of-the-art performance, on classification tasks for 3D object recognition benchmark datasets.

## 1    Introduction

Rapid advances in 3D sensing technology have made 3D data ubiquitous and easily accessible, rendering them an important data source for high level semantic understanding in a variety of environments. The semantic understanding problem, however, remains very challenging for 3D data as it is hard to find an effective scheme for converting input data into informative features for further processing by machine learning algorithms. For semantic understanding problems in 2D images, deep CNNs [15] have been widely

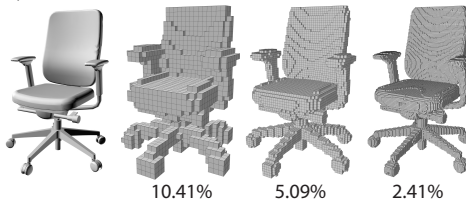

10.41%          5.09%          2.41%

Figure 1: The sparsity characteristic of 3D data in occupancy grid representation. 3D occupancy grids in resolution 30, 64 and 128 are shown in this figure, together with their density, defined as $\frac{\#occupied\ grid}{\#total\ grid}$. It is clear that 3D occupancy grid space gets sparser and sparser as the fidelity of the surface approximation increases.

used and have achieved great success, where the convolutional layers play an essential role. They provide a set of 2D filters, which when convolved with input data, transform the data to informative features for higher level inference.

In this paper, we focus on the problem of learning a 3D shape representation by a deep neural network. We keep two goals in mind when designing the network: the shape features should be *discriminative* for shape recognition and *efficient* for extraction at runtime. However, existing 3D CNN pipelines that simply replace the conventional 2D filters by 3D ones [31, 19], have difficulty in capturing geometric structures with sufficient efficiency. The input to these 3D CNNs are voxelized shapes represented by occupancy grids, in direct analogy to pixel array representation for images. We observe that the computational cost of 3D convolution is quite high, since convolving 3D voxels has cubical complexity with respect to spatial resolution, one order higher than the 2D case. Due to this high computational cost, researchers typically choose $30 \times 30 \times 30$ resolution to voxelize shapes [31, 19], which is significantly lower than the widely adopted resolution $227 \times 227$ for processing images [24]. We suspect that the strong artifacts introduced at this level of quantization (see Figure 1) hinder the process of learning effective 3D convolutional filters.

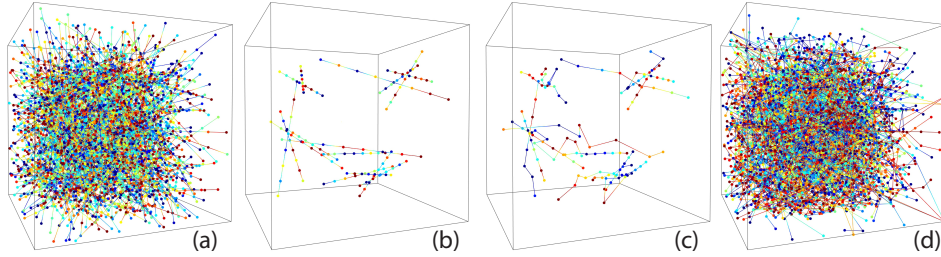

Figure 2: An visualization of probing filters before (a) and after (d) training them for extracting 3D features. The colors associated with each probing point visualize the filter weights for them. Note that probing points belong to the same filter are linked together for visualizing purpose. (b) and (c) are subsets of probing filters of (a) and (d), for better visualizing that not only the weights on the probing points, but also their locations are optimized for them to better "sense" the space.

Two significant differences between 2D images and 3D shapes interfere with the success of directly applying 2D CNNs on 3D data. First, *as the voxel resolution grows, the grids occupied by shape surfaces get sparser and sparser* (see Figure 1). The convolutional layers that are designed for 2D images thereby waste much computation resource in such a setting, since they convolve with 3D blocks that are largely empty and a large portion of multiplications are with zeros. Moreover, *as the voxel resolution grows, the local 3D blocks become less and less discriminative*. To capture informative features, long range connections have to be established for taking distant voxels into consideration. This long range effect demands larger 3D filters, which yields an even higher computation overhead.

To address these issues, we represent 3D data as 3D fields, and propose a *field probing* scheme, which samples the input field by a set of *probing filters* (see Figure 2). Each probing filter is composed of a set of probing points which determine the shape and location of the filter, and filter weights associated with probing points. In typical CNNs, only the filter weights are trained, while the filter shape themselves are fixed. In our framework, due to the usage of 3D field representation, both the weights and probing point locations are trainable, making the filters highly flexible in coupling long range effects and adapting to the sparsity of 3D data when it comes to feature extraction. The computation amount of our field probing scheme is determined by how many probing filters we place in the 3D space, and how many probing points are sampled per filter. Thus, the computational complexity does not grow as a function of the input resolution. We found that a small set of field probing filters is enough for sampling sufficient information, probably due to the sparsity characteristic of 3D data.

Intuitively, we can think our field probing scheme as a set of sensors placed in the space to collect informative signals for high level semantic tasks. With the long range connections between the sensors, global overview of the underlying object can be easily established for effective inference. Moreover, the sensors are "smart" in the sense that they learn how to sense the space (by optimizing the filter weights), as well as where to sense (by optimizing the probing point locations). Note that the intelligence of the sensors is not hand-crafted, but solely derived from data. We evaluate our field probing based neural networks (FPNN) on a classification task on ModelNet [31] dataset, and show that they match the performance of 3DCNNs while requiring much less computation, as they are designed and trained to respect the sparsity of 3D data.

## 2   Related Work

**3D Shape Descriptors.**   3D shape descriptors lie at the core of shape analysis and a large variety of shape descriptors have been designed in the past few decades. 3D shapes can be converted into 2D images and represented by descriptors of the converted images [13, 4]. 3D shapes can also be represented by their inherent statistical properties, such as distance distribution [22] and spherical harmonic decomposition [14]. Heat kernel signatures extract shape descriptions by simulating an heat diffusion process on 3D shapes [29, 3]. In contrast, we propose an approach for learning the shape descriptor extraction scheme, rather than hand-crafting it.

**Convolutional Neural Networks.**   The architecture of CNN [15] is designed to take advantage of the 2D structure of an input image (or other 2D input such as a speech signal), and CNNs have advanced the performance records in most image understanding tasks in computer vision [24]. An important reason for this success is that by leveraging large image datasets (e.g., ImageNet [6]), general purpose image descriptors can be directly learned from data, which adapt to the data better and outperform hand-crafted features [16]. Our approach follows this paradigm of feature learning, but is specifically designed for 3D data coming from object surface representations.

**CNNs on Depth and 3D Data.** With rapid advances in 3D sensing technology, depth has became available as an additional information channel beyond color. Such 2.5D data can be represented as multiple channel images, and processed by 2D CNNs [26, 10, 8]. Wu et al. [31] in a pioneering paper proposed to extend 2D CNNs to process 3D data directly (3D ShapeNets). A similar approach (VoxNet) was proposed in [19]. However, such approaches cannot work on high resolution 3D data, as the computational complexity is a cubic function of the voxel grid resolution. Since CNNs for images have been extensively studied, 3D shapes can be rendered into 2D images, and be represented by the CNN features of the images [25, 28], which, surprisingly, outperforms any 3D CNN approaches, in a 3D shape classification task. Recently, Qi et al. [23] presented an extensive study of these volumetric and multi-view CNNs and refreshed the performance records. In this work, we propose a feature learning approach that is specifically designed to take advantage of the sparsity of 3D data, and compare against results reported in [23]. Note that our method was designed without explicit consideration of deformable objects, which is a purely extrinsic construction. While 3D data is represented as meshes, neural networks can benefit from intrinsic constructions[17, 18, 1, 2] to learn object invariance to isometries, thus require less training data for handling deformable objects.

Our method can be viewed as an efficient scheme of sparse coding[7]. The learned weights of each probing curve can be interpreted as the entries of the coding matrix in the sparse coding framework. Compared with conventional sparse coding, our framework is not only computationally more tractable, but also enables an end-to-end learning system.

# 3 Field Probing Neural Network

## 3.1 Input 3D Fields

We study the 3D shape classification problem by employing a deep neural network. The input of our network is a 3D vector field built from the input shape and the output is an object category label. 3D shapes represented as meshes or point clouds can be converted into 3D distance fields. Given a mesh (or point cloud), we first convert it into a binary occupancy grid representation, where the binary occupancy value in each grid is determined by whether it intersects with any mesh surface (or contains any sample point). Then we treat the occupied cells as the zero level set of a surface, and apply a distance transform

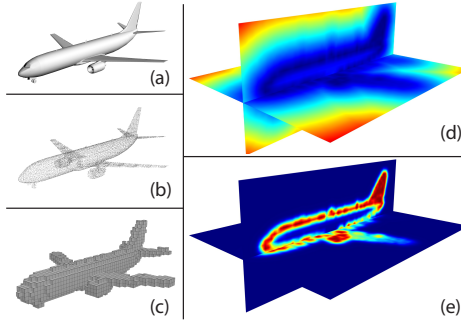

Figure 3: 3D mesh (a) or point cloud (b) can be converted into occupancy grid (c), from which the input to our algorithm — a 3D distance field (d), is obtained via a distance transform. We further transform it to a Gaussian distance field (e) for focusing attention to the space near the surface. The fields are visualized by two crossing slices.

to build a 3D distance field $\mathcal{D}$, which is stored in a 3D array indexed by $(i, j, k)$, where $i, j, k = 1, 2, ..., R$, and $R$ is the resolution of the distance field. We denote the distance value at $(i, j, k)$ by $\mathcal{D}_{(i,j,k)}$. Note that $\mathcal{D}$ represents distance values at discrete grid locations. The distance value at an arbitrary location $d(x, y, z)$ can be computed by standard trilinear interpolation over $\mathcal{D}$. See Figure 3 for an illustration of the 3D data representations.

Similar to 3D distance fields, other 3D fields, such as normal fields $\mathcal{N}_x$, $\mathcal{N}_y$, and $\mathcal{N}_z$, can also be used for representing shapes. Note that the normal fields can be derived from the gradient of the distance field: $\mathcal{N}_x(x, y, z) = \frac{1}{l}\frac{\partial d}{\partial x}, \mathcal{N}_y(x, y, z) = \frac{1}{l}\frac{\partial d}{\partial y}, \mathcal{N}_z(x, y, z) = \frac{1}{l}\frac{\partial d}{\partial z}$, where $l = |(\frac{\partial d}{\partial x}, \frac{\partial d}{\partial y}, \frac{\partial d}{\partial z})|$. Our framework can employ any set of fields as input, as long as the gradients can be computed.

## 3.2 Field Probing Layers

The basic modules of deep neural networks are layers, which gradually convert input to output in a *forward pass*, and get updated during a *backward pass* through the *Back-propagation* [30] mechanism. The key contribution of our approach is that we replace the convolutional layers in CNNs by field probing layers, a novel component that uses field probing filters to efficiently extract features from the 3D vector field. They are composed of three layers: *Sensor layer*,

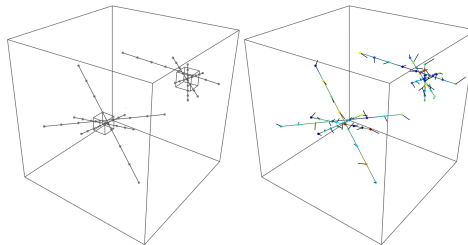

Figure 4: Initialization of field probing layers. For simplicity, a subset of the filters are visualized.

*DotProduct layer* and *Gaussian layer*. The Sensor layer is responsible for collecting the signals (the values in the input fields) at the probing points in the forward pass, and updating the probing point locations in the backward pass. The DotProduct layer computes the dot product between the probing filter weights and the signals from the Sensor layer. The Gaussian layer is an utility layer that transforms distance field into a representation that is more friendly for numerical computation. We introduce them in the following paragraphs, and show that they fit well for training a deep network.

**Sensor Layer.** The input to this layer is a 3D field $\mathcal{V}$, where $\mathcal{V}(x, y, z)$ yields a $T$ channel ($T = 1$ for distance field and $T = 3$ for normal fields) vector at location $(x, y, z)$. This layer contains $C$ probing filters scattered in space, each with $N$ probing points. The parameters of this layer are the locations of all probing points $\{(x_{c,n}, y_{c,n}, z_{c,n})\}$, where $c$ indexes the filter and $n$ indexes the probing point within each filter. This layer simply outputs the vector at the probing points $\mathcal{V}(x_{c,n}, y_{c,n}, z_{c,n})$. The output of this layer forms a data chunk of size $C \times N \times T$.

The gradient of this function $\nabla \mathcal{V} = (\frac{\partial \mathcal{V}}{\partial x}, \frac{\partial p}{\partial y}, \frac{\partial p}{\partial z})$ can be evaluated by numerical computation, which will be used for updating the locations of probing points in the back-propagation process. This formal definition emphasizes why we need the input being represented as 3D fields: the gradients computed from the input fields are the forces to push the probing points towards more informative locations until they converge to a local optimum.

**DotProduct Layer.** The input to this layer is the output of the Sensor layer — a data chunk of size $C \times N \times T$, denoted as $\{p_{c,n,t}\}$. The parameters of DotProduct layer are the filter weights associated with probing points, i.e., there are $C$ filters, each of length $N$, in $T$ channels. We denote the set of parameters as $\{w_{c,n,t}\}$. The function at this layer computes a dot product between $\{p_{c,n,t}\}$ and $\{w_{c,n,t}\}$, and outputs $v_c = v(\{p_{c,i,j}\}, \{w_{c,i,j}\}) = \sum_{\substack{i=1,\ldots,N \\ j=1,\ldots,T}} p_{c,i,j} \times w_{c,i,j}$, — a $C$-dimensional vector, and the gradient for the backward pass is: $\nabla v_c = (\frac{\partial v}{\partial \{p_{c,i,j}\}}, \frac{\partial v}{\partial \{w_{c,i,j}\}}) = (\{w_{c,i,j}\}, \{p_{c,i,j}\})$.

Typical convolution encourages weight sharing within an image patch by "zipping" the patch into a single value for upper layers by a dot production between the patch and a 2D filter. Our DotProduct layer shares the same "zipping" idea, which facilitates to fully connect it: probing points are grouped into probing filters to generate output with lower dimensionality.

Another option in designing convolutional layers is to decide whether their weights should be shared across different spatial locations. In 2D CNNs, these parameters are usually shared when processing general images. In our case, we opt not to share the weights, as information is not evenly distributed in 3D space, and we encourage our probing filters to individually deviate for adapting to the data.

**Gaussian Layer.** Samples in locations distant to the object surface are associated with large distance values from the distance field. Directly feeding them into the DotProduct layer does not converge and thus does not yield reasonable performance. To emphasize the importance of samples in the vicinity of the object surface, we apply a Gaussian transform (inverse exponential) on the distances so that regions approaching the zero surface have larger weights while distant regions matter less.[1]. We implement this transform with a Gaussian layer. The input is the output values of the Sensor layer. Let us assume the values are $\{x\}$, then this layer applies an element-wise Gaussian transform $g(x) = e^{-\frac{x^2}{2\sigma^2}}$, and the gradient is $\nabla g = -\frac{xe^{-\frac{x^2}{2\sigma^2}}}{\sigma^2}$ for the backward pass.

**Complexity of Field Probing Layers.** The complexity of field probing layers is $O(C \times N \times T)$, where $C$ is the number of probing filters, $N$ is the number of probing points on each filter, and $T$ is the number of input fields. The complexity of the convolutional layer is $O(K^3 \times C \times S^3)$, where $K$ is the 3D kernel size, $C$ is the output channel number, and $S$ is the number of the sliding locations for each dimension. In field probing layers, we typically use $C = 1024$, $N = 8$, and $T = 4$ (distance and normal fields), while in 3D CNN $K = 6$, $C = 48$ and $S = 12$. Compared with convolutional layers, field probing layers save a majority of computation ($1024 \times 8 \times 4 \approx 1.83\% \times 6^3 \times 48 \times 12^3$), as the

probing filters in field probing layers are capable of learning where to "sense", whereas convolutional layers exhaustively examine everywhere by sliding the 3D kernels.

**Initialization of Field Probing Layers.** There are two sets of parameters: the probing point locations and the weights associated with them. To encourage the probing points to explore as many potential locations as possible, we initialize them to be widely distributed in the input fields. We first divide the space into $G \times G \times G$ grids and then generate $P$ filters in each grid. Each filter is initialized as a line segment with a random orientation, a random length in $[l_{low}, l_{high}]$ (we use $[l_{low}, l_{high}] = [0.2, 0.8] * R$ by default), and a random center point within the grid it belongs to (Figure 4 left). Note that a probing filter spans distantly in the 3D space, so they capture long range effects well. This is a property that distinguishes our design from those convolutional layers, as they have to increase the kernel size to capture long range effects, at the cost of increased complexity. The weights of field probing filters are initialized by the Xavier scheme [9]. In Figure 4 right, weights for distance field are visualized by probing point colors and weights for normal fields by arrows attached to each probing point.

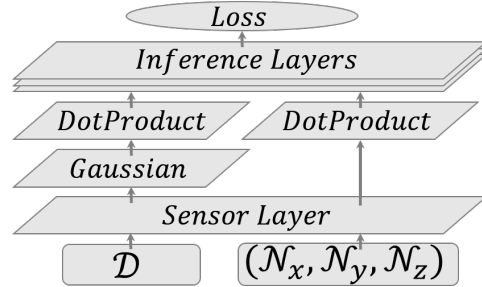

**FPNN Architecture and Usage.** Field probing layers transform input 3D fields into an intermediate representation, which can further be processed and eventually linked to task specific loss layers (Figure 5). To further encourage long range connections, we feed the output of our field probing layers into fully connected layers. The advantage of long range connections makes it possible to stick with a small number of probing filters, while the small number of probing filters makes it possible to directly use fully connected layers.

Figure 5: FPNN architecture. Field probing layers can be used together with other inference layers to minimize task specific losses.

Object classification is widely used in computer vision as a testbed for evaluating neural network designs, and the neural network parameters learned from this task may be transferred to other high-level understanding tasks such as object retrieval and scene parsing. Thus we choose 3D object classification as the task for evaluating our FPNN.

## 4 Results and Discussions

### 4.1 Timing

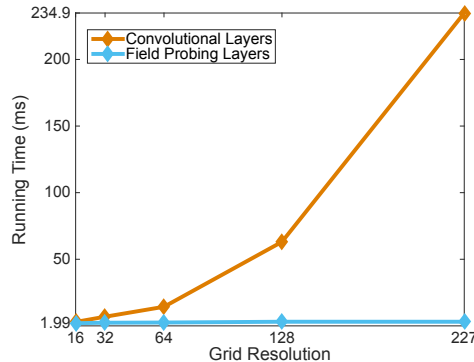

We implemented our field probing layers in Caffe [12]. The Sensor layer is parallelized by assigning computation on each probing point to one GPU thread, and DotProduct layer by assigning computation on each probing filter to one GPU thread. Figure 6 shows a run time comparison between convolutional layers and field probing layers on different input resolutions. The computation cost of our field probing layers is agnostic to input resolutions, the slight increase of the run time on higher resolution is due to GPU memory latency introduced by the larger 3D fields. Note that the convolutional layers

Figure 6: Running time of convolutional layers (same settings as that in [31]) and field probing layers ($C \times N \times T = 1024 \times 8 \times 4$) on Nvidia GTX TITAN with batch size $8^3$.

in [12] are based on highly optimized cuBlas library from NVIDIA, while our field probing layers are implemented with our naive parallelism, which is likely to be further improved.

### 4.2 Datasets and Evaluation Protocols

We use ModelNet40 [31] (12,311 models from 40 categories, training/testing split with 9,843/2,468 models[4]) — the standard benchmark for 3D object classification task, in our experiments. Models

in this dataset are already aligned with a canonical orientation. For 3D object recognition scenarios in real world, the gravity direction can often be captured by the sensor, but the horizontal "facing" direction of the objects are unknown. We augment ModelNet40 data by randomly rotating the shapes horizontally. Note that this is done for both training and testing samples, thus in the testing phase, the orientation of the inputs are unknown. This allows us to assess how well the trained network perform on real world data.

### 4.3 Performance of Field Probing Layers

| 1-FC | | | 4-FCs | | |
|---|---|---|---|---|---|
| w/o FP | w/ FP | +NF | w/o FP | w/ FP | +NF |
| 79.1 | 85.0 | 86.0 | 86.6 | 87.5 | 88.4 |

Table 1: Top-1 accuracy of FPNNs on 3D object classification task on $ModelNet40$ dataset.

We train our FPNN $80,000$ iterations on $64 \times 64 \times 64$ distance field with batch size $1024$.[5], with SGD solver, learning rate $0.01$, momentum $0.9$, and weight decay $0.0005$.

Trying to study the performance of our field probing layers separately, we build up an FPNN with only one fully connected layer that converts the output of field probing layers into the representation for softmax classification loss (1-FC setting). Batch normalization [11] and rectified-linear unit [20] are used in-between our field probing layers and the fully connected layer for reducing internal covariate shift and introducing non-linearity. We train the network without/with updating the field probing layer parameters. We show their top-1 accuracy on 3D object classification task on $ModelNet40$ dataset with single testing view in Table 1. It is clear that our field probing layers learned to sense the input field more intelligently, with a $5.9\%$ performance gain from $79.1\%$ to $85.0\%$. Note that, what achieved by this simple network, $85.0\%$, is already better than the state-of-the-art 3DCNN before [23] ($83.0\%$ in [31] and $83.8\%$ in [19]).

We also evaluate the performance of our field probing layers in the context of a deeper FPNN, where four fully connected layers[6], with in-between batch normalization, rectified-linear unit and Dropout [27] layers, are used (4-FCs setting). As shown in Table 1, the deeper FPNN performs better, while the gap between with and without field probing layers, $87.5\% - 86.6\% = 0.9\%$, is smaller than that in one fully connected FPNN setting. This is not surprising, as the additional fully connected layers, with many parameters introduced, have strong learning capability. The $0.9\%$ performance gap introduced by our field probing layers is a precious extra over a strong baseline.

It is important to note that in both settings (1-FC and 4-FCs), our FPNNs provides reasonable performance even without optimizing the field probing layers. This confirms that long range connections among the sensors are beneficial.

Furthermore, we evaluate our FPNNs with multiple input fields (+NF setting). We did not only employ distance fields, but also normal fields for our probing layers and found a consistent performance gain for both of the aforementioned FPNNs (see Table 1). Since normal fields are derived from distance fields, the same group of probing filters are used for both fields. Employing multiple fields in the field probing layers with different groups of filters potentially enables even higher performance.

**Robustness Against Spatial Perturbations.** We evaluate our FPNNs on different levels of spatial perturbations, and summarize the results in Table 2, where $R$ indicates random horizontal

| FPNN Setting | $R$ | $R_{15}$ | $R_{15} + T_{0.1} + S$ | $R_{45}$ | $T_{0.2}$ |
|---|---|---|---|---|---|
| 1-FC | 85.0 | 82.4 | 76.2 | 74.1 | 72.2 |
| 4-FCs | 87.5 | 86.8 | 84.9 | 85.3 | 85.4 |
| [31] | 84.7 | - | - | 83.0 | 84.8 |

Table 2: Performance on different perturbations.

rotation, $R_{15}$ indicates $R$ plus a small random rotation $(-15°, 15°)$ in the other two directions, $T_{0.1}$ indicates random translations within range $(-0.1, 0.1)$ of the object size in all directions, $S$ indicates random scaling within range $(0.9, 1.1)$ in all directions. $R_{45}$ and $T_{0.2}$ shares the same notations, but with even stronger rotation and translation, and are used in [23] for evaluating the performance of [31]. Note that such perturbations are done on both training and testing samples. It is clear that our FPNNs are robust against spatial perturbations.

**Advantage of Long Range Connections.** We evaluate our FPNNs with different range parameters $[l_{low}, l_{high}]$ used in initializing the probing filters, and summarize the results in Table 3. Note that since the output dimensionality

| FPNN Setting | $0.2 - 0.8$ | $0.2 - 0.4$ | $0.1 - 0.2$ |
|---|---|---|---|
| 1-FC | 85.0 | 84.1 | 82.8 |
| 4-FCs | 87.5 | 86.8 | 86.9 |

Table 3: Performance with different filter spans.

of our field probing layers is low enough to be directly feed into fully connected layers, distant sensor information is directly coupled by them. This is a desirable property, however, it poses the difficulty to study the advantage of field probing layers in coupling long range information separately. Table 3 shows that even if the following fully connected layer has the capability to couple distance information, the long range connections introduced in our field probing layers are beneficial.

**Performance on Different Field Resolutions.**
We evaluate our FPNNs on different input field resolutions, and summarize the results in Table 4. Higher resolution input fields can represent in-

| FPNN Setting | $16 \times 16 \times 16$ | $32 \times 32 \times 32$ | $64 \times 64 \times 64$ |
|---|---|---|---|
| 1-FC | 84.2 | 84.5 | 85.0 |
| 4-FCs | 87.3 | 87.3 | 87.5 |

Table 4: Performance on different field resolutions.

put data more accurately, and Table 4 shows that our FPNN can take advantage of the more accurate representations. Since the computation cost of our field probing layers is agnostic to the resolution of the data representation, higher resolution input fields are preferred for better performance, while coupling with efficient data structures reduces the I/O footprint.

**"Sharpness" of Gaussian Layer.** The $\sigma$ hyper-parameter in Gaussian layer controls how "sharp" is the transform. We select its value empirically in our experiments, and the best performance is given when we use $\sigma \approx 10\%$ of the object size. Smaller $\sigma$ slightly hurts the performance ($\approx 1\%$), but has the potential of reducing I/O footprint.

**FPNN Features and Visual Similarity.** Figure 7 shows a visualization of the features extracted by the FPNN trained for a classification task. Our FPNN is capable of capturing 3D geometric structures such that it allows to map 3D models that belong to the same categories (indicated by colors) to similar regions in the feature space. More specifically, our FPNN maps 3D models into points in a high dimensional feature space, where the distances between the points measure the similarity between their corresponding 3D models. As can be seen from Figure 7 (better viewed in zoomin mode), the FPNN feature distances be-

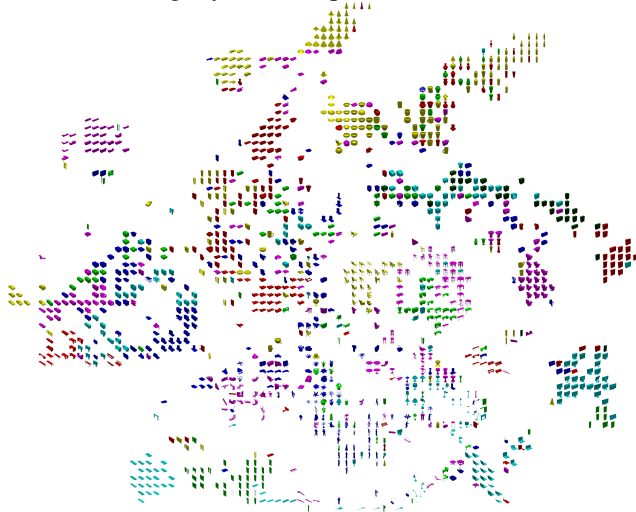

Figure 7: t-SNE visualization of FPNN features.

tween 3D models represent their shape similarities, thus FPNN features can support shape exploration and retrieval tasks.

### 4.4 Generalizability of FPNN Features

| Testing Dataset | FP+FC | FC Only | FP+FC on Source FC Only on Target |
|---|---|---|---|
| $MN40_1$ | 93.8 | 90.7 | 92.7 |
| $MN40_2$ | 89.4 | 85.1 | 88.2 |

Table 5: Generalizability test of FPNN features.

One superior characteristic of CNN features is that features from one task or dataset can be transferred to another task or dataset. We evaluate the generalizability of FPNN features by cross validation — we train on one dataset and test on another. We first split $ModelNet40$ (lexicographically by the category names) into two parts $MN40_1$ and $MN40_2$, where each of them contains 20 non-overlapping categories. Then we train two FPNNs in a 1-FC setting (updating both field probing layers and the only one fully connected layer) on these two datasets, achieving 93.8% and 89.4% accuracy, respectively (the second column in Table 5).[7] Finally, we fine tune only the fully connected layer of these two FPNNs on the dataset that they were not trained from, and achieved 92.7% and 88.2% on $MN40_1$ and $MN40_2$, respectively (the fourth column in Table 5), which is comparable to that directly trained from the testing categories. We also trained two FPNNs in 1-FC setting with updating only the fully connected layer, which achieves 90.7% and 85.1% accuracy on $MN40_1$ and $MN40_2$, respectively (the third column in Table 5). These two FPNNs do not perform as well as the fine-tuned FPNNs (90.7% < 92.7% on $MN40_1$

and $85.1\% < 88.2\%$ on $MN40_2$), although all of them only update the fully connected layer. These experiments show that the field probing filters learned from one dataset can be applied to another one.

## 4.5 Comparison with State-of-the-art

| Our FPNN | [23] | |
|---|---|---|
| (4-FCs+NF) | SubvolSup+BN | MVCNN-MultiRes |
| 88.4 | 88.8 | 93.8 |

Table 6: Comparison with state-of-the-art methods.

We compare the performance of our FPNNs against two state-of-the-art approaches — SubvolSup+BN and MVCNN-MultiRes, both from [23], in Table 6. SubvolSup+BN is a subvolume supervised volumetric 3D CNN, with batch normalization applied during the training, and MVCNN-MultiRes is a multi-view multi-resolution image based 2D CNN. Note that our FPNN achieves comparable performance to SubvolSup+BN with less computational complexity. However, both our FPNN and SubvolSup+BN do not perform as well as MVCNN-MultiRes. It is intriguing to answer the question why methods directly operating on 3D data cannot match or outperform multi-view 2D CNNs. The research on closing the gap between these modalities can lead to a deeper understanding of both 2D images and 3D shapes or even higher dimensional data.

## 4.6 Limitations and Future Work

**FPNN on Generic Fields.** Our framework provides a general means for optimizing probing locations in 3D fields where the gradients can be computed. We suspect this capability might be particularly important for analyzing 3D data with invisible internal structures. Moreover, our approach can easily be extended into higher dimensional fields, where a careful storage design of the input fields is important for making the I/O footprint tractable though.

**From Probing Filters to Probing Network.** In our current framework, the probing filters are independent to each other, which means, they do not share locations and weights, which may result in too many parameters for small training sets. On the other hand, fully shared weights greatly limit the representation power of the probing filters. A trade-off might be learning a probing network, where each probing point belongs to multiple "pathes" in the network for partially sharing parameters.

**FPNN for Finer Shape Understanding.** Our current approach is superior for extracting robust global descriptions of the input data, but lacks the capability of understanding finer structures inside the input data. This capability might be realized by strategically initializing the probing filters hierarchically, and jointly optimizing filters at different hierarchies.

## 5 Conclusions

We proposed a novel design for feature extraction from 3D data, whose computation cost is agnostic to the resolution of data representation. A significant advantage of our design is that long range interaction can be easily coupled. As 3D data is becoming more accessible, we believe that our method will stimulate more work on feature learning from 3D data. We open-source our code at `https://github.com/yangyanli/FPNN` for encouraging future developments.

### Acknowledgments

We would first like to thank all the reviewers for their valuable comments and suggestions. Yangyan thanks Daniel Cohen-Or and Zhenhua Wang for their insightful proofreading. The work was supported in part by NSF grants DMS-1546206 and IIS-1528025, UCB MURI grant N00014-13-1-0341, Chinese National 973 Program (2015CB352501), the Stanford AI Lab-Toyota Center for Artificial Intelligence Research, the Max Planck Center for Visual Computing and Communication, and a Google Focused Research award.

## Footnotes

[1]Applying a batch normalization [11] on the distances also resolves the problem. However, Gaussian transform has two advantages: 1. it can be approximated by truncated distance fields [5], which is widely used in real time scanning and can be compactly stored by voxel hashing [21], 2. it is more efficient to compute than batch normalization, since it is element-wise operation.

[3]The batch size is chosen to make sure the largest resolution data fits well in GPU memory.

[4]The split is provided on the authors' website. In their paper, a split composed of at most 80/20 training/testing models for each category was used, which is tiny for deep learning tasks and thus prone to overfitting. Therefore, we report and compare our performance on the whole ModelNet40 dataset.

[5]To save disk I/O footprint, a data augmentation is done on the fly. Each iteration, 256 data samples are loaded, and augmented into 1024 samples for a batch.

[6]The first three of them output 1024 dimensional feature vector.

[7]The performance is higher than that on all the 40 categories, since the classification task is simpler on less categories. The performance gap between $MN40_1$ and $MN40_2$ is presumably due to the fact that $MN40_1$ categories are easier to classify than $MN40_2$ ones.

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
