[Reviews · NeurIPS 2016]

Reviewer 1 Summary This paper introduces a 3D CNN model with field-probing layers in place of the convolutional layers in CNNs. The presentation is clear and detailed. Although only one 3D object dataset with 40 classes was employed in the experiments, the study on the robustness, feature extraction and generalization abilities was carried out sensibly and gave some promising results. The limitations of the work were clearly outlined and future directions suggested. Even though the proposed network did not beat the multi-view 2D CNNs in recognition performance, the work seems meaningful and informative to future research. Qualitative Assessment The paper is in general well written with admirable clarity and particularity. The setting of sigma for the Gaussian layer did not seem to be given. What is the effect of changing its value? I suspect by having a larger sigma would allow the model to &quot;see through&quot; and model inner contours (holes e.g.), which is potentially a nice capability, especially when this is impossible with multiple view 2D CNNs. However, the success of the 2D CNN models may also suggest that the surface areas are most decisive in object recognition, which means you should prefer small values for sigma? It&#39;ll be nice to have more discussion in this regard. Confidence in this Review 1-Less confident (might not have understood significant parts) Reviewer 2 Summary The paper introduces Field Probing Neural Networks, an extrinsic construction based on 3D volumetric fields that circumvents limitations of voxel based approaches. Qualitative Assessment The paper introduces Field Probing Neural Networks, an extrinsic construction based on 3D volumetric fields that circumvents limitations of voxel based approaches. The paper is well written and I find the idea rather interesting, despite not having a huge gap in raw performance (but a huge one in terms of computational resources). A minor note regarding readability. There are many repetitions (mostly nouns) in the text which could be removed to make it easier to read. &quot;However, existing 3D CNN pipelines&quot; -> I would remove However. Figure 1: An visualization -> A visualization. I would like the authors to make clear that their construction is purely extrinsic and that therefore in case of deformable objects it will not be invariant to isometries. This means that much larger training sets will be needed as the network has to learn this invariance from the data itself. In case of intrinsic constructions this would be build in in the representation of the 3D space. Lines 131:132. This sentence needs to be rewritten. Gradually can be misleading. Deep nets define a highly non-linear hierarchical input to output mapping. Parameters of such layers, if any (max-pooling has none), are usually optimized via gradient descent using the back-propagation algorithm to compute the gradient. However, one could use other optimization algorithm for instance (evolutionary etc). I have a remark on the Gaussian Layer. As written in the note this is mainly a normalization issue, and I don&#39;t see why batch normalization is such a big efficiency problem. One could argue that it is important to keep the right level of sparsity (focus on the object surface) and that the gaussian layer helps in doing that. Have the authors experimented with learnable variances (\delta), I think this will be the final step to have a fully end-to-end system. Lines 210:213. In case of deformable shapes I argue that training will be much harder, reason for which intrinsic deep learning is being investigated in [1,2,3]. Regarding the experiments. The distance field used as input is of size 64x64x64, I think that in order to be completely fair with the 3DShapeNet work an experiment with 30x30x30 should be performed as well. Of course the advantage of FPNN is that they can be scaled much better, on this note have the authors experimented with larger field sizes? I think this would give insights on what is the right level of detail needed for this classification task and that was not possibly to explore with more naive methods. Line 357:358. You optimize for location, therefore it may happen that after training some filters do converge to the same location. Is it the case? Have the authors investigated these type of redundancies in the parameterization? Line 360. Pathes should read paths? Source code will be released, this is a big plus for reproducibility. Missing references: [1] LITMAN R., BRONSTEIN A. M.: Learning spectral descriptors for deformable shape correspondence. [2] MASCI J., BOSCAINI D., BRONSTEIN M. M., VAN DERGHEYNST P.: Shapenet: Convolutional neural networks on non-euclidean manifolds. [3] BOSCAINI D., MASCI J., RODOLÀ E., BRONSTEIN M. M., CREMERS D.: Anisotropic diffusion descriptors. Confidence in this Review 3-Expert (read the paper in detail, know the area, quite certain of my opinion) Reviewer 3 Summary This paper focuses on the problem of object classification based on 3D data that one could acquire with the use of some sort of range sensor. The basic idea is that for 3D data one would ideally like to have 3D filters in a convolutional neural network. However, 3D filters are quite expensive to be optimized and used, and they are pretty wasteful since the majority of the input will probably be zeros (only near the object surface there are non-zero values). As a note here, the input is a 3D rectangle split into a 3D grid, where the original object point cloud is converted in a simplified, binarized form (you can think of it as building the surface of an object using LEGO blocks). As such the paper proposes a field probing layer which replaces the convolutional layers with a combination of three "sub-layers": a) a sensor layer, b) a dot-product layer and c) a gaussian layer. The sensor layer learns to place sampling points/probes on the 3D rectangle space, which would be equivalent to learning the sampling/convolutional locations in a ConvNet. The gaussian layer implements a gaussian function that smoothens the sampling points from the sensor layer. And the dot-product is a standard inner product layer to compute the inner product between the sampling layer and some learnt weights. Interestingly, the weights are not shared in the field probing layer, meaning that in different locations different "probing filters" are learnt. A strong positive point is that the proposed approach can learn complex interaction between surface points, even if they are not neighboring ones. The reason is that the sensor layers are not constrained to place the probes within a confined space. Qualitative Assessment The paper is quite interesting, although not entirely novel or delivering impressive results. The list of positive and negative points is the following. Positive points ------------------- + The problem is interesting and an approach that could deliver some good results would definitely have some impact. Indeed, relying on filters with cubic complexity for 3DCNN is unnecessarily intense, since most of the input space is composed of zeros. Learning more &quot;surface-focused&quot; filters make much more sense. + Modelling long-range surface dependencies is sensible, especially when scaling up to datasets with hundreds of categories. + The presentation of the paper is very good and the method is very clear. Negative points ------------------- - In its current form a field probing can be viewed as some sort of sparse tensor encoding. As such it is not clear what is the novelty as compared to having a sparse encoding of the input space and the weight filters. This is also kind of supported by figure 5, where indeed the proposed method has a straight line (the same one would expect by a sparse encoding), whereas a 3D convolutional network naturally feature a cubic complexity. One difference I can identify is the fact that long-range dependencies can be learnt, namely that the the probing locations are learnt instead of just defined. Is that a difference and is it the only main difference? - It is not clear how can you learn meaningful long-range dependencies within such an unconstrained setting. With normal convolutional filters there exists a very strong constraint, namely the inputs should be nearby. In your case how can you make sure the network does not get stuck in a poor local optimum? - The fact that the weights in the field probing layers are not shared means that they are tied to the particular locations of the input. How generalizable is that? When you are presented with a random point cloud for which you cannot know the rotation as compared to the ground truth, how could these filters cope with the variance? - Extending on the point, how do you perform the rotation changes in Table 2? Do you apply the same, fixed rotation to all objects or do you rotate randomly per test object? If it is the former, does the experiment bring any insight, as I assume that the classification should be more or less similar for any random (but common across examples) viewpoint. If it is the latter, what is the difference between R and R_15? - Also, since the weights are not shared, does the network generalize for objects of different sizes, since in that case one would naturally need to increase the granularity so that to be able to maintain the appopriate level of detail for the recognition? If this cannot be guaranteed, doesn&#39;t this go against the point of having a more efficient network so that one can increase the grid resolution on demand? Overall I believe the paper is quite interesting and although by itself does not excel, it could serve as a potential inspiration to other researchers. Confidence in this Review 2-Confident (read it all; understood it all reasonably well) Reviewer 4 Summary From reading the paper it seems the dot product layer and the gaussian layers are parallel in precedence yet sometimes it feels one comes before another. No flowchart of the model also makes it hard to understand the concept. The initialization of the weights could be better explained too. The probing weights Vs the probing length concept needs clear explanation. Finally the results section doesn't say much as it doesn't talk with reference to any standard network or established results. Qualitative Assessment It seems that the text lacks continuity. While it seems a promising work in terms of novelty and speeding up 3D classifications it seems the lack of a flowchart for the layer algorithm makes the text very confusing. It becomes hard to understand what exactly is going on inside the Field probing layers and in what sequence. How exactly is this layer replaceable compared to a standard CNN all these pieces needs more explanation and sound proof. The code provided is not very well documented either. While it can be understood how one may use the code on the test side but no utils is provided for training instructions, like do you intend to replace the CNN layers in the standard Caffe models of Alexnet/ Googlenet? if so you need to provide code samples of how exactly it can be done like a train. prototxt file. For the result section some evaluation is run on the ModelNet40 dataset but no comparison to an existing process that uses a CNN based architecture is given. Confidence in this Review 2-Confident (read it all; understood it all reasonably well) Reviewer 5 Summary This paper presents a convolutional neural network (CNN) for 3D data classification task. As input, the method considers 3D distance fields converted from meshes or point clouds. Instead of 3D convolution used in some previous work, they propose the Gaussian layer to map a distance field to Gaussian distance field, the sensor layer for probing (sampling) 3D voxel space, and the dot product layer for weighting the probed data vector, which allow efficient operation. The proposed layers are used at the lowest layer followed by fully connected (FC) layers. The key difference with previous CNN is that locations of sensing points at the sensor layer are learned from data during training. This can be viewed as a special case or an extension of [R1]. [R1] Jaderberg et al., Spatial Transformer Networks, NIPS 2015. Qualitative Assessment =Pros= 1. The proposed network is more efficient than 3DCNN with comparable or less accuracy than other state-of-the-art methods. 2. The sensor probing layer is somewhat novel (even it can be viewed as a special case of STN). 3. The algorithms are evaluated in various aspects (but limited). =Cons= 1. Some evaluations are conducted with only limited regimes. 2. Only empirical analysis. After discussing with other reviewers, the concerns on the evaluation was resolved, and the reviewer raised my score on this paper. (In the final version, clearly rewriting the footnote 4 is suggested. See the fatal flaw section.) For technical quality, the score is given due to the evaluations on the limited regimes. The evaluation in terms of various aspects is admitted, but very scarcely sampled control parameters prevents to deduce and predict the behavior of the algorithm. Also, once the model were evaluated on more than a single dataset, it convinces the reviewer and it truly improve the quality of the paper. There are several 3D datasets. Among the following datasets, [1] has comparable number of data to ModelNet40. [1] ICRA 2011, http://cs.stanford.edu/people/teichman/stc/ [2] AAAI 2010, http://www2.informatik.uni-freiburg.de/~spinello/pcloud-dataset.html [3] ACRA 2013, http://www.acfr.usyd.edu.au/papers/SydneyUrbanObjectsDataset.shtml For the novelty, still for this reviewer, the sensor probe layer can be considered as a special case of Spatial Transformer Networks (STN) [1]. The probe layer can be viewed as a simple STN model with simply changing the parameterizations of the STN layer originally from region sampling to point sampling and from affine model to translation model. The authors do not admit this point, but I would recommend to check this point again. Confidence in this Review 3-Expert (read the paper in detail, know the area, quite certain of my opinion)